# Ebola Entry Inhibitors Discovered from *Maesa perlarius*

**DOI:** 10.3390/ijms23052620

**Published:** 2022-02-27

**Authors:** Nga Yi Tsang, Wan-Fei Li, Elizabeth Varhegyi, Lijun Rong, Hong-Jie Zhang

**Affiliations:** 1Teaching and Research Division, School of Chinese Medicine, Hong Kong Baptist University, Kowloon, Hong Kong, China; 12012173@life.hkbu.edu.hk (N.Y.T.); wanfeier@live.cn (W.-F.L.); 2Department of Microbiology and Immunology, College of Medicine, University of Illinois Chicago, 909 South Wolcott Ave, Chicago, IL 60612, USA; estark2434@gmail.com

**Keywords:** anti-Ebola virus activity, *Maesa perlarius*, procyanidin, condensed tannin, flavan-3-ol, EGCG

## Abstract

Ebola virus disease (EVD), a disease caused by infection with Ebola virus (EBOV), is characterized by hemorrhagic fever and a high case fatality rate. With limited options for the treatment of EVD, anti-Ebola viral therapeutics need to be urgently developed. In this study, over 500 extracts of medicinal plants collected in the Lingnan region were tested against infection with Ebola-virus-pseudotyped particles (EBOVpp), leading to the discovery of *Maesa perlarius* as an anti-EBOV plant lead. The methanol extract (MPBE) of the stems of this plant showed an inhibitory effect against EBOVpp, with an IC_50_ value of 0.52 µg/mL, which was confirmed by testing the extract against infectious EBOV in a biosafety level 4 laboratory. The bioassay-guided fractionation of MPBE resulted in three proanthocyanidins (procyanidin B2 (**1**), procyanidin C1 (**2**), and epicatechin-(4*β*→8)-epicatechin-(4*β*→8)-epicatechin-(4β→8)-epicatechin (**3**)), along with two flavan-3-ols ((+)-catechin (**4**) and (−)-epicatechin (**5**)). The IC_50_ values of the compounds against pseudovirion-bearing EBOV-GP ranged from 0.83 to 36.0 µM, with **1** as the most potent inhibitor. The anti-EBOV activities of five synthetic derivatives together with six commercially available analogues, including EGCG ((−)-epigallocatechin-3-*O*-gallate (**8**)), were further investigated. Molecular docking analysis and binding affinity measurement suggested the EBOV glycoprotein could be a potential molecular target for **1** and its related compounds.

## 1. Introduction

Ebola virus (EBOV) is the causative agent of Ebola virus disease (EVD), a disease associated with hemorrhagic fever and high case fatality rates [1,2,3]. With limited therapeutic options and a current outbreak of Ebola virus disease in Africa, there is an urgent need to develop novel anti-EBOV agents [4]. The study of anti-EBOV agents is largely hampered by the requirement of biosafety level 4 (BSL-4) containments. 

In the current study, to search for inhibitors of Ebola virus, we used a cell-based assay with replication-incompetent pseudotyped viral particles that express a specific viral glycoprotein on the surface and can be performed in a BSL-2 containment [5]. The development of the “one-stone-two-birds” protocol was first reported in 2011 [6]. This approach has been subsequently applied in our plant drug discovery program and assisted in our successful discovery of a number of antiviral lead compounds [5,7,8,9,10].

Over 50% of approved drugs are natural products or their derivatives and mimics, which illustrates that plants are an important source for drug discovery [11]. There are nearly 33,000 vascular plant species found in China, and 9300 seed plant species are used in traditional Chinese medicine [12]. Lingnan refers to the tropical and subtropical region to the south of the Nan Mountains, a mountain range that divides the south and central subtropical zones [13]. The numerous plant resources with wide biological diversity in this region provide enormous potential for the discovery of antiviral agents.

A collection of over 500 medicinal plant extracts from the Lingnan region was tested against infection with Ebola-virus-pseudotyped HIV particles (EBOVpp). This led to the identification of *Maesa perlarius* as an anti-EBOV plant lead, which was further investigated to discover active compounds through bioassay-guided separation.

*M*. *perlarius* is a 1–3 m tall shrub mainly distributed in China, Thailand, and Vietnam [14,15]. *M*. *perlarius* has several ethnopharmacological uses in different regions. In China, *M. perlarius* is known as *ji yu dan*. The leaves of *M*. *perlarius* are made into paste for the treatment of broken bones. In Cambodia, Laos, and Vietnam, the roots are used as diuretics and digestives. The leaves are used as a remedy for treating measles, and an infusion of the leaves can be prepared as a drink for postpartum care [15]. The phytochemistry of *Maesa* species has been extensively studied, with triterpenoid saponins being the most well studied, followed by flavonoids and benzoquinones [16,17,18,19,20,21]. However, compounds with antiviral properties have rarely been reported from *Maesa* plants. 

In our bioassay-guided fractionation study of the methanol extract of the plant *M*. *perlarius*, we isolated a number of active polyphenols composed of flavan-3-ol units, including procyanidins (also referred to as “proanthocyanidins”). Dimeric procyanidins and several flavan-3-ol monomers were found to be potent entry inhibitors against EBOV, with IC_50_ values ranging from 0.83 to 2.95 µM. The subsequent mechanism of action and binding affinity measurement, aided by docking analysis and microscale thermophoresis (MST) technology, determined that the flavan-3-ol compounds exhibited virucidal potency by interacting with EBOV glycoprotein. In the present study, we report on structural isolation, determination, and modification and evaluate the anti-EBOV activity of the identified polyphenol compounds.

## 2. Results

### 2.1. Stem Extract of M. perlarius Is Identified as a Potential Anti-EBOV Lead

The screening of Lingnan plant extracts led to the identification of *M. perlarius* (MPBE) as an anti-EBOV hit plant. The dose–response relationship of the MPBE extract was established, and its IC_50_ value against infection with replication-incompetent EBOVpp was measured as 0.52 µg/mL (Figure 1). MPBE was further evaluated and validated using infectious Ebola virus in a biosafety level 4 (BSL-4) containment facility, which revealed an inhibitory effect against EBOV, with an IC_50_ value of 7.47 µg/mL.

### 2.2. Flavan-3-Ols and Procyanidins Identified from the Stems of M. perlarius

The methanol extract (MPBE, 529 g) of the stems of *M. perlarius* (5.5 kg) was fractionated by silica-gel normal-phase column chromatography with the use of a gradient elution system of dichloromethane (CH_2_Cl_2_)/acetone, followed by the gradient CH_2_Cl_2_/MeOH, to afford 84 fractions (MPBF1–84). MPBF64 (48.8 g), one of the eluents of CH_2_Cl_2_:MeOH 1:1, was found to exhibit an inhibitory effect against replication-incompetent EBOVpp, with an IC_50_ value of 0.42 µg/mL. A portion of this fraction (4.1 g) was then subjected to further separation by a reverse-phase column packed with octadecylsilyl (ODS) silica gel and eluted by gradient MeOH/H_2_O to afford 11 fractions (MPBF64A-F64K) (Figure 2). The obtained fractions were evaluated for their anti-EBOV activity. Most fractions separated from F64 (except F64J and F64K) showed more than 50% inhibition of EBOVpp luciferase signal compared to the luciferase signal of VSVpp (vesicular stomatitis virus (VSV)-pseudotyped HIV particles) at a concentration of 1.0 µg/mL. There was no cytotoxicity against the host cells at a concentration of 20 µg/ mL, confirming that the inhibitory effects of these fractions are not due to cytotoxicity but because it is acting on the binding and entry stage of EBOVpp. Fractions of MPBF64C–G showed negative inhibitory effects towards VSVpp, suggesting that the fractions might contain components that promote VSVpp infection. The active fraction MPBF64B (0.65 g) was further purified by a Sephadex LH-20 column and preparative HPLC to provide five compounds. By comparing the spectral data with those of previously identified compounds, they were identified as the known compounds procyanidins B2 (**1**) and C1 (**2**), epicatechin-(4β→8)-epicatechin-(4β→8)-epicatechin-(4β→8)-epicatechin (**3**), (+)-catechin (**4**), and (−)-epicatechin (**5**) (Figure 3).

### 2.3. Anti-EBOV Activity Evaluation of Proanthocyanidins and Flavan-3-Ols

For a better comparison of the compounds’ activities, we further investigated the dose–response relationship of their anti-EBOV activities, together with some commercially available B-type procyanidin and flavan-3-ols (Figure 4). The IC_50_ values of the compounds against EBOVpp ranged from 0.83 to 36.0 µM, and compound **1** was found to be the most potent (Table 1). All of the B-type procyanidins (**1**, **6**, and **7**) exhibited potent anti-EBOV activities, with IC_50_ values ranging from 0.83 to 1.52 µM. Both compounds **1** and **6** possessed 2*R*, 3*R*, and 4*R* configurations on the upper unit. The difference in the configuration of C-2′ and C-3′ at the terminal unit is not crucial for the anti-EBOV activity of compounds **1** and **6**. Compounds **6** and **7** have different stereochemistry on their upper units (**6**: 2*R*, 3*R*, and 4*R*; **7**: 2*R*, 3*S,* and 4*S*), which contributes to the stronger anti-EBOV activity of **6** (IC_50_ = 0.95 µM) over **7** (IC_50_ = 1.52 µM). This anti-EBOV activity and difference in structural relationship can also be observed in gallocatechin (**10**) and epigallocatechin (**11**). With a 2*R* and 3*R* stereochemistry, compound **11** showed 2.2 times stronger antiviral activity than compound **10** (2*R* and 3*S*).

Catechin (**4**) and epicatechin (**5**) are the basic units used by nature to form proanthocyanidins, but the two monomers displayed much weaker anti-EBOVpp activity than the dimeric proanthocyanidins. However, the ester analogues, EGCG (**8**) and ECG (**9**), displayed much stronger anti-EBOV activities than compounds **4** and **5**. EGCG was also reported as an inhibitor of the endoplasmic reticulum (ER) chaperone heat shock 70 kDa protein 5 (HSPA5), which could inhibit EBOV infection [22]. Another study applied an in-silico analysis to dock a series of polyphenol compounds to Ebola viral protein VP24, which showed that EGCG and procyanidin B2 had binding affinity to VP24 [23].

Compounds **4, 5**, **10**, and **11** are flavan-3-ol monomers. With one more hydroxyl group substituted on the ring B, the anti-EBOV activity of compounds **10** and **11** were ~3–4 times stronger than that of compounds **4** and **5**, respectively.

To extend the scope of the study, a series of additional flavan-3-ol analogues were screened for their inhibitory effects against EBOVpp (Appendix A). Surprisingly, none of them exhibited an anti-EBOV effect, even at concentrations of 50 µM. This suggests that the flavan-3-ol skeleton is crucial for the anti-EBOV activity of these compounds.

### 2.4. Synthesis of ECG Analogues from (−)-Epicatechin to Study the Functional Group Requirement for Anti-EBOV Activity

Since ECG (**9**) showed strong anti-EBOV activity (Table 1) and considering its structural simplicity in comparison with EGCG, it was selected as a structural scaffold to synthesize the ester analogues at C-3 and further explore the anti-EBOV activity enhancement potential. (−)-Epicatechin (**5**) was then used as the base structure for structural modification due to its commercial availability. The derivatization of compound **5** started from the selective benzyl (Bn) protection on its phenol group by reacting with benzyl bromide and potassium carbonate (K_2_CO_3_) in super-dry DMF overnight. Following column separation to purify the 3′,4′,5,7-tetra-*O*-benzyl-(–)-epicatechin (**12**), the hydroxyl group at C-3 underwent Steglich esterification with various benzoyl chlorides, carbonyl chlorides, benzoic acids, and carboxylic acids in the presence of 4-dimethylaminopyridine (4-DMAP) to provide epicatechin derivatives with a structural similarity to EGCG. To improve the reactivity of benzoic acid and carboxylic acid towards esterification, these acids were reacted with oxalyl chloride to form the relevant acyl chlorides. After esterification, the hydroxyl groups were deprotected by hydrogenolysis to yield the final products. Three benzoic acid esters (**13b**–**15b**) and two cycloalkylcarboxylate esters (**16b** and **17b**) were synthesized (Figure 1). 

The cytotoxicity and antiviral activity of these synthetic compounds were tested at 10 µM. None of the synthetic compounds exhibited improved anti-EBOV activity, and only **13b** exhibited > 60% inhibitory effect against EBOVpp infection, which suggests that the 3,4,5-trihydroxybenzoate ester at C-3 of (–)-epicatechin could be crucial for anti-EBOV activity (Figure 5).

### 2.5. Molecular Docking Analysis and Microscale Thermophoresis (MST) Measurement Reveal EBOV-GP as the Potential Molecular Target of B-Type Procyanidins and Flavan-3-Ol Analogues

In recent years, computational approaches have been widely used to elucidate the binding targets of bioactive compounds at the molecular level. The characterization of the EBOV virion structure and the availability of co-crystalized 3D structures of the Ebola viral glycoprotein (EBOV-GP) with ligands provides a basis for molecular docking analysis. EBOV-GP is a single protein located at the outer membrane of EBOV and plays an important role in viral attachment, endosomal entry, and fusion. EBOV-GP possesses a trimeric structure, which is composed of the external segment, GP1, and the transmembrane unit, GP2 [24]. Up to now, 11 compounds have been found to have direct interaction with EBOV-GP. These chemically diversified structures were shown to bind at the same cavity of EBOV-GP but with varied binding modes [25,26,27,28]. It has been suggested that binding of these compounds to this large cavity of EBOV-GP could destabilize the protein and compromise EBOV entry [26]. A computational approach was thus employed to explore the possibility of this EBOV-GP binding cavity as the binding site of B-type procyanidins and their related compounds.

To evaluate whether EBOV-GP is the potential target of B-type procyanidins and their related compounds, procyanidins B1 (**6**) and B2 (**1**) were docked with the binding domains of EBOV-GP crystal structures in complex with the reported anti-EBOV compounds using AutoDock Vina (Figure 6) [27]. These compounds could be docked to the binding pocket of toremifene (the known ligand of EBOV-GP) in the EBOV-GP crystal structure 5JQ7 [25,29]. The docking condition was then further applied to the docking of other compounds.

We docked eight B-type proanthocyanidins or the flavan-3-ols monomers to the EBOV-GP binding cavity (Appendix A). The correlation between the docking score and the IC_50_ value of a ligand was also studied. As shown in Table 2, the docking scores and IC_50_ values were inversely correlated with r = −0.919 (Pearson correlation coefficient, *p* < 0.01). There is a trend that a compound with a lower IC_50_ value has a lower binding score, indicating that the compound may have a higher binding affinity to the cavity. The molecular docking results suggested that compounds **1**, and **6** and their related compounds possibly bind to this Ebola GP cavity to destabilize the Ebola viral protein, which suggests that EBOV-GP is the potential target protein for procyanidin B2 and its related compounds.

To confirm the molecular interaction of B-type procyanidins with EBOV-GP, MST experiments were performed to measure the binding affinity of EBOV-GP and procyanidin B2 (**6**). MST is a qualitative method to elucidate the interaction between a protein and a ligand by measuring the diffusion behavior of the labeled protein when the protein is being excited by infrared light. The diffusion behavior of the protein is altered later, when it is complexed with an increased number of unlabeled ligand molecules. The binding affinity (*K*_d_) of the ligand can be determined by obtaining the titration curves [31]. By definition, the dissociation constant (*K*_d_) refers to the concentration of ligand at which half of the ligand binding sites on the protein are occupied by the ligand molecules [32].

The His-tag labeling strategy was employed to characterize the interaction between our protein of interest and a small molecule in purification-free cell lysate [33]. In this experiment, a cell lysate expressing EBOV-GP-His_6_ was used to determine binding affinity. A pCMV vector coding for only His_6_ was used as a control. We determined the EBOV-GP *K*_d_ value for procyanidin B2 binding to EBOV-GP to be 13.0 µM, whereas the *K*_d_ value for toremifene (positive control) was measured as 21.4 µM (Figure 7 and Appendix A), which is similar to values reported in the literature (24.3 µM and 14 µM) using different assay buffer systems [25,34]. On the other hand, toremifene and procyanidin B2 could not produce a binding curve with the negative control (Appendix A).

## 3. Discussion

Ebola virus disease (EVD) is a highly pathogenic disease with a case death rate of more than 60%. In the present study, a pseudotyped virus-based screening assay system was established to discover specific EBOV inhibitors against EBOV-GP-mediated entry, which includes attachment, endocytosis, and fusion [6,24]. The inhibitors may target viral proteins or host factors that are involved in viral infection. The screening of 500 medicinal plants from the Lingnan region of China resulted in the identification of *M. perlarius* as an anti-EBOV plant lead. The methanol extract (MPBE) of the stem materials of this plant was subjected to column separation guided by our developed anti-Ebola pseudoviral system, leading to isolation of three procyanidins (procyanidin B2 (**1**), procyanidin C1 (**2**), and epicatechin-(4*β*→8)-epicatechin-(4*β*→8)-epicatechin-(4β→8)-epicatechin (**3**)), along with two flavan-3-ol monomers ((+)-catechin (**4**) and (−)-epicatechin (**5**)). To the best of our knowledge, this is the first isolation and identification of procyanidins from the *Maesa* species. 

Procyanidins are condensed tannins that are formed through the C-C linkage of flavan-3-ol monomers, i.e., catechin and epicatechin. However, the biochemical pathway of procyanidin synthesis is not fully understood. Similar to their monomers, procyanidins are polyphenolic compounds and are present in many edible plants. Procyanidins are abundant in unripe apples, barley, sorghum, peanuts, cacao, various berries, and grapes [35]. They were found to have antioxidative effects, and some procyanidin-rich plant extracts have been commercially available in the market as dietary supplements [36]. For example, Pycnogenol, the standardized extract of French maritime pine bark, was found to contain over 70% procyanidin [37]. The 3D structure of a procyanidin largely depends on the stereochemistry of the hydroxyl group at C-3 and the bonds formed between the monomers. Procyanidins with different spatial conformations contribute to the differences in interaction with molecules and ligands, resulting in different bioactivities [38]. Besides antioxidative effects, the immunomodulatory, cardioprotective, and platelet-activation effects of procyanidins were also reported [39]. Condensed tannins and their monomers were reported to exhibit antiviral activities against various kinds of viruses, including HSV, HIV, SARS-CoV, and H1N1 influenza virus [40,41,42,43], and our study is the first report of in vitro anti-EBOV viral activities of procyanidins. 

In the current study, the anti-EBOV activity of proanthocyanidins and flavan-3-ols were investigated. The IC_50_ values of these compounds against infection with EBOVpp in A549 cells ranged from 0.83 µM to 36.0 µM. Compound **1** was the most potent compound, with an IC_50_ value of 0.83 µM. B-type proanthocyanidins showed stronger overall anti-EBOV activity than their flavan-3-ol monomers. Configurations of the functional groups in proanthocyanidins also affect antiviral activity. Since proanthocyanidins and flavan-3-ols belong to large classes of phytochemicals, there are ample opportunities to explore the structure–activity relationship in order to identify anti-EBOV lead molecules among polyphenols. 

Understanding of the anti-EBOV mechanism of procyanidins is very limited. Their related flavan-3-ol monomer compounds, gallic acid (IC_50_ value of 25.4 µM against infectious recombinant EBOV infection) and EGCG, were previously reported as EBOV entry inhibitors. The time-of-addition assay revealed that gallic acid targeted a post-binding step and possibly interferes with the GP-mediated fusion. EGCG was found to be an inhibitor of ER chaperone HSPA5, an Ebola virus associated host protein [22,44].

In our study, we attempted to investigate EBOV-GP as the possible molecular target of procyanidin B2 and its related compounds by using molecular docking analysis in conjunction with MST technology to verify the binding affinity of the ligand–protein interaction. The docking scores were calculated to approximate the binding energies of the flavan-3-ol derivatives to the protein target, EBOV-GP. Our docking analysis results indicated that the structures of these flavan-3-ol derivatives could fit into a known EBOV-GP binding cavity targeted by multiple chemically diverse drugs. Though the scores might not be equivalent to the IC_50_s [45], a good correlation was found by comparing the docking scores and the experimental in vitro IC_50_ values of the eight investigated flavan-3-ol compounds (Table 2).

The molecular docking results were further validated by MST experiments to measure the EBOV-GP binding affinities of procyanidin B2 in comparison to toremifene. Different from surface plasmon resonance (SPR) technology, which detects change in size after the ligand-target binding event, the measurement of the thermophoretic profile in MST is achieved by observing tiny changes in the solvation entropy of the target molecule. MST has several advantages, such as sample preparation and method development. It does not require immobilization of a target protein to a surface or require a complicated buffer system. Purification-free samples can also be used, which allows for rapid determination of binding affinity of ligand–protein interaction [46]. Our MST experiments confirmed that procyanidin B2 is a ligand of EBOV-GP, with a *K*_d_ value of 13.0 μM, revealing that this compound binds to the same cavity of EBOV-GP as toremifene (*K*_d_ value of toremifene was measured at 21.4 μM).

Taking these results into account, EBOV-GP could be the correct antiviral target for these flavan-3-ol molecules, and the anti-EBOV potency of these types of compounds could be further improved by manipulating the flavan-3-ol molecule structures to fit in the procyanidin B2 binding site with enhanced binding affinities.

## 4. Materials and Methods

### 4.1. General

Normal-phase column chromatography was performed using silica gel (100–230 mesh, Davisil or 230–400 mesh, Merck, Darmstadt, Germany). A Sephadex LH-20 (Amersham BioSciences UK Ltd., Amersham, UK) was used in size-exclusion chromatography. Reversed-phase column chromatography was carried out on ODS RP-18 silica gel (40–63 μm, Merck). Analytical high-performance liquid chromatography (HPLC) was carried on an Agilent Technologies Series 1100 HPLC (Agilent Technologies, Inc., Santa Clara, CA, USA) with a diode-array detector (DAD). An analytical Thermo-C18 (150 × 4.6 mm) column was used in the analysis. Semi-preparative HPLC was performed on the same HPLC system equipped with a semi-preparative column YMC-Pack ODS-A, S-5 μm, 12 nm (250 × 20 mm) (YMC Co., Ltd., Kyoto, Japan). The high-resolution electrospray ionization mass spectroscopy (HRESIMS) spectra were acquired from an Agilent 6540 time-of-flight mass spectrometer (Agilent Technologies). The optical rotations were measured with a JASCO P-1010 polarimeter (JASCO, Easton, MD, USA). The circular dichroism (CD) spectra were acquired on an Olis DSM 172 circular dichroism spectrophotometer (Olis, Athens, GA, USA). The infrared (IR) spectra were recorded on a PerkinElmer Spectrum One FT-IR spectrometer using potassium bromide (KBr) pellets (PerkinElmer, Waltham, MA, USA). The ultraviolet–visible (UV-vis) spectra were obtained from a Varian Cary 50 UV-Vis spectrophotometer (Agilent Technologies, Santa Clara, CA, USA). The 1D and 2D nuclear magnetic resonance (NMR) spectra were acquired with a Bruker Ascend 400 MHz spectrometer (Bruker, Karlsruhe, Germany). The unit of chemical shifts (*δ*) in NMR spectra are presented in parts per million (ppm). The chemical shifts of deuterated methanol (CD_3_OD) were used as references (^1^H: 3.31 ppm, ^13^C: 49.00 pm). All coupling constants (*J*) were expressed in Hertz (Hz). The standard pulse sequence provided by the manufacturer was applied in all NMR experiments.

Procyanidins B1 (purity ≥ 95%) and B3 (purity ≥ 95%) were purchased from Biopurify (Chengdu, China). The chromanes mentioned in Section 2.3 were purchased from TargetMol (Boston, MA, USA). (+) Catechin (purity ≥ 98%), (−) epicatechin (purity ≥ 98%), gallocatechin (purity ≥ 98%), epigallocatechin (purity ≥ 98%), epicatechin 3-*O*-gallate (purity ≥ 98%), and epigallocatechin 3-O-gallate (purity ≥ 98%) were purchased from Sigma-Aldrich (St. Louis, MO, USA).

The chemicals and reagents used in the synthesis were purchased from AnalaR NORMAPUR (VWR Chemicals, Radnor, PA, USA), J&K (J&K Scientific, Beijing, China), Sigma-Aldrich (Sigma-Aldrich Corporation, St. Louis, MO, USA), Labscan (RCI Labscan Limited, Bangkok, Thailand), Dieckmann (Dieckmann Company, Shenzhen, China), and TCI (TCI Development Co., Shanghai, China).

### 4.2. Plant Materials

The stems of *M**. perlarius* were collected in Hong Kong in October 2014. The collected plants were authenticated by Prof. Chen Hu-Biao of the School of Chinese Medicine, Hong Kong Baptist University. A voucher specimen (SHB0005) available for inspection upon request at the Quality Research Laboratory/Phytochemistry Laboratory, School of Chinese Medicine, Hong Kong Baptist University. The plant materials were allowed to dry at room temperature for two weeks and were then cut into small pieces and ground into powder by a pulverizer.

### 4.3. Extraction and Isolation

The dried powder of the stems of *M**. perlarius* (5.5 kg) was extracted by methanol (3 × 32 L, 24 h) at room temperature. The extract was filtered out and concentrated to dryness by rotary evaporators to afford an extract (MPBE, 529 g), which was fractionated by silica-gel normal-phase column chromatography (1.5 kg, 100–230 mesh, Davisil™) with the use of elution system dichloromethane (CH_2_Cl_2_)/acetone (100:0; 90:10; 80:20; 70:30), followed by gradient CH_2_Cl_2_/MeOH (95:5; 90:10; 85:15; 80:20; 70:30; 50:50; 0:100), to afford 84 fractions (MPBF1-84). MPBF64 (48.8 g), one of the fractions eluted by CH_2_Cl_2_:MeOH 1:1, was found to exhibit an inhibitory effect against Ebola virus. A portion of this fraction (4.1 g) was therefore subjected to further separation by a reverse-phase column packed with ODS (40 g, 100–200 mesh, Chromatorex^®^; Fuji Silysia Chemical Ltd., Kasugai, Japan), eluting by gradient MeOH/H_2_O (90:10; 70:30; 50:50; 30:70; 0:100) to afford 11 fractions (MPBF64A-F64K). 

MPBF64B (0.65 g) was further separated by a Sephadex LH-20 column using MeOH as an eluting solvent to afford 151 fractions (MPBFB1-MPBFB151). Fractions MPBFB16-MPBFB17 (27.85 mg), MPBFB19 (27 mg), and MPBFB20 (30 mg) were purified by preparative HPLC (Appendix A) on a C18 column (Saphir™ 110 C18, 12 μm, 300 × 40 mm) using gradient MeCN/H_2_O mobile phase to provide compounds **1** (5.18 mg), **2** (8.52 mg), **3** (5.93 mg), **4** (0.75 mg), and **5** (2.51 mg).

Procyanidin B2 (**1**) was isolated as a pale-yellow amorphous powder, ${\left[\alpha \right]}_{D}^{25}$ +26.7 (*c* 0.07, MeOH). UV (MeOH) *λ*_max_ (log ε) 205 (4.52), 280 (3.72) nm. IR (KBr) *ν*_max_ 3400, 1615, 1525, 1449, 1384, 1286, 1111 cm^−1^. **^1^**H NMR (400 MHz, CD_3_OD) *δ* 7.11 (brs, 1H), 6.88 (d, *J* = 2.2 Hz, 1H), 6.73 (d, *J* = 7.7 Hz, 2H), 6.70 (dd, *J* = 7.7, 2.2 Hz, 2H), 5.94 (brs, 3H), 5.04 (brs, 1H), 4.95 (brs, 1H), 4.63 (brs, 1H), 4.28 (brs, 1H), 3.90 (brs, 1H), 2.91 (dd, *J* = 15.5, 3.5 Hz, 1H), 2.81 (d, *J* = 2.9 Hz, 1H) ppm. HRESIMS *m/z* 579.1503 [M + H]^+^ (calculated for C_30_H_2__6_O_12_, 579.1503). The chemical shifts and the number of proton signals in the ^1^H NMR spectrum of compounds **1** matched the ^1^H NMR spectrum of procyanidin B2 reported by Makabe et al. [47] (Appendix A). The retention time, ion peak, and fragmentation pattern of compound **1** in LC-MS also matched with those of the purchased procyanidin B2 standard (Chengdu Biopurify Phytochemicals Ltd., Chengdu, China). Compound **1** was identified as procyanidin B2.

Procyanidin C1 (**2**) was isolated as a pale-yellow amorphous powder, ${\left[\alpha \right]}_{D}^{25}$ +55.6 (*c* 0.10, MeOH). UV (MeOH) *λ*_max_ (log ε) 200 (4.64), 280 (3.51) nm. IR (KBr) *ν*_max_ 3400, 1615, 1526, 1448, 1286, 1114 cm^−1^. ^1^H NMR (400MHz, CD_3_OD) *δ* 7.13 (brs, 1H), 7.03 (brs, 1H), 6.93 (d, *J* = 8.1 Hz, 1H), 6.92 (d, *J* = 2.0 Hz, 1H), 6.77 (dd, *J* = 8.1, 2.0 Hz, 2H), 6.74 (d, *J* = 8.0 Hz, 2H), 6.71 (dd, *J* = 8.1, 2.1 Hz, 1H), 6.04 (brs, 1H), 6.01 (d, *J* = 1.9 Hz, 1H), 5.93 (s, 2H), 5.22 (brs, 1H), 5.08 (brs, 1H), 5.00 (brs, 1H), 4.71 (d, *J* = 8.8 Hz, 1H), 4.60 (brs, 1H), 4.33 (brs, 1H), 4.01 (d, *J* = 8.1 Hz, 1H), 2.94 (dd, *J* = 16.0, 4.3 Hz, 1H), 2.83 (d, *J* = 16.0 Hz, 1H) ppm. HRESIMS *m/z* 867.2089 [M + H]^+^ (calculated for C_45_H_38_O_18_, 867.2136). The spectroscopic data matched those of procyanidin C1 reported by Saito et al. [48] (Appendix A). Therefore, compound **2** was identified as procyanidin C1.

Epicatechin-(4β→8)-epicatechin-(4β→8)-epicatechin-(4β→8)-epicatechin (**3**) was isolated as a pale-yellow amorphous powder, ${\left[\alpha \right]}_{D}^{25}$ +107.2 (*c* 0.09, MeOH). UV (MeOH) *λ*_max_ (log ε) 195 (4.48), 280 (3.55) nm. IR (KBr) *ν*_max_ 3400, 1614, 1525, 1447, 1384, 1286, 1204, 1114 cm^−1^. ^1^H NMR (400 MHz, CD_3_OD) *δ* 7.14 (brs, 1H), 7.09 (brs, 1H), 7.04 (brs, 1H), 6.93 (brs, 2H), 6.78 (dd, *J* = 8.3, 2.3 Hz, 3H), 6.74 (d, *J* = 8.3 Hz, 3H), 6.71 (d, *J* = 8.3 Hz, 1H), 6.05 (brs, 1H), 6.01 (brs, 1H), 5.94 (brs, 3H), 5.26 (brs, 2H), 5.09 (brs, 1H), 5.01 (brs, 1H), 4.79 (brs, 1H), 4.74 (brs, 2H), 4.34 (brs, 1H), 4.12 (brs, 1H), 4.03 (brs, 2H), 2.95 (dd, *J* = 17.0, 3.7 Hz, 1H), 2.84 (d, *J* = 17.0 Hz, 1H) ppm. HRESIMS *m/z* 1155.2749 [M + H]^+^ (calculated for C_60_H_50_O_24_, 1155.2770). The ^1^H NMR and MS data of compound **3** matched those of a procyanidin tetramer reported by Shoji et al. (Appendix A) [49]. Compound **3** was determined as epicatechin-(4*β*→8)-epicatechin-(4*β*→8)-epicatechin-(4*β*→8)-epicatechin.

(+)-Catechin (**4**) was isolated as a white amorphous powder, ${\left[\alpha \right]}_{D}^{25}$ +14.1 (*c* 0.07, MeOH). UV (MeOH) *λ*_max_ (log ε) 195 (4.75), 280 (3.73) nm. IR (KBr) *ν*_max_ 3368, 1627, 1520, 1469, 1373, 1284, 1143 cm^−1^. HRESIMS *m/z* 291.0862 [M + H]^+^ (calculated for C_15_H_14_O_6_, 281.0869). The retention time, ion peak, and fragmentation pattern of compound **4** in LC-MS matched those of the purchased (+)-catechin standard (Sigma-Aldrich, St. Louis, MO, USA). Accordingly, compound **4** was identified as (+)-catechin.

(−)-Epicatechin (**5**) was isolated as a white amorphous powder, ${\left[\alpha \right]}_{D}^{25}$ −54.8 (*c* 0.12, MeOH). UV (MeOH) *λ*_max_ (log ε) 210 (4.74), 280 (3.56) nm. IR (KBr) *ν*_max_ 3456, 2930, 1627, 1521, 1469, 1262, 1145 cm^−1^. HRESIMS *m/z* 291.0861 [M + H]^+^ (calculated for C_15_H_14_O_6_, 291.0869). The retention time, ion peak, and fragmentation pattern of **5** in LC-MS matched those of purchased (−)-epicatechin (Sigma-Aldrich, St. Louis, MO, USA). Compound **5** was therefore identified as (−)-epicatechin.

### 4.4. Synthesis of ECG Derivatives

The starting compound, (−)-epi-catechin, was purchased from Tokyo Chemical Industry (TCI, Shanghai, China). All other reagents were purchased from Sigma-Aldrich or J&K (Beijing, China). (−)-Epi-catechin was first selectively protected by Bn-groups as described in the literature [50]. The protected epi-catechin (12) was then carried out to react with benzoyl chloride, carbonyl chloride, benzoic acid, and carboxylic acid in the presence of 4-DMAP in CH_2_Cl_2_ to afford the derivatives. After esterification, the hydroxyl groups were deprotected by hydrogenolysis to afford the final products. The spectra for these compounds are provided in the supporting information.

(2*R*,3*R*)-5,7-Bis(benzyloxy)-2-(3,4-bis(benzyloxy)phenyl)chroman-3-ol (**12**) was obtained as a white solid [50]. Following the addition of potassium carbonate (K_2_CO_3_) (6 mmol) into the solution of (−)-epicatechin (1 mmol) in dimethylformamide (DMF) (5.0 mL), benzyl bromide (4.2 mmol) was added dropwisely at room temperature under N_2_ protection. After stirring at room temperature for 20 h, the reaction mixture was quenched with water (6 mL). The product was partitioned with ethyl acetate (EtOAc). The EtOAc layer was washed with water and dried with anhydrous sodium sulphate (Na_2_SO_4_). The product was purified by flash-column chromatography (n-Hex/EA 7:1) to afford a white solid (45%). HRESIMS 651.2536 *m/z* [M + H]^+^ (calculated for C_43_H_38_O_6_, 651.2747). ^1^H NMR (400 MHz, CDCl_3_) δ 7.45–7.29 (m, 20H, Ar-H), 7.13 (d, *J* = 1.9 Hz, 1H, 2′-H), 6.98 (d, *J* = 1.9, 1H, 6′-H), 6.95 (d, *J* = 8.3 Hz, 1H, 5′-H), 6.26 (d, *J* = 2.3 Hz, 1H, 6-H), 6.25 (d, *J* = 2.3 Hz, 1H, 8-H), 5.17 (d, *J* = 5.9 Hz, 4H, CH_2_Ph), 5.00 (d, *J* = 4.0 Hz, 4H, CH_2_Ph), 4.90 (s, 1H, 2-H), 4.20 (s, 1H, 3-H), 3.03–2.95 (m, 1H, 4α-H), 2.91 (dd, *J* = 17.3, 4.4 Hz, 1H, 4β-H) ppm.

(2*R*,3*R*)-5,7-Bis(benzyloxy)-2-(3,4-bis(benzyloxy)phenyl)chroman-3-yl benzoate (**13a**) was obtained as a yellow oily liquid. Compound **12** (0.07 mmol) and 4-DMAP (0.175 mmol) in CH_2_Cl_2_ (dichloromethane) (4 mL) were added dropwisely to a solution of benzoyl chloride (0.14 mmol) in CH_2_Cl_2_ (1 mL) at 0 °C. The reaction mixture was stirred at room temperature for 16 h. Saturated sodium bicarbonate (NaHCO_3_) solution was added into the reaction mixture, and the organic layer was separated. The aqueous layer was further extracted with CH_2_Cl_2_. The combined CH_2_Cl_2_ layers were then dried over Na_2_SO_4_. The dried and evaporated residue was purified over PTLC (n-Hex/EtOAc 3:1) to afford the desired compound **13a** as a yellow oily liquid (60%). HRESIMS 755.2950 *m/z* [M + H]^+^ (calculated for C_50_H_42_O_7_, 755.3008). ^1^H NMR (CDCl_3_, 400 MHz) δ 8.02–7.94 (2H, m, Ar-H), 7.42–7.28 (22H, m, Ar-H), 7.19 (1H, d, *J* = 2.0 Hz, 10-H), 6.99 (1H, dd, *J* = 8.4, 2.0 Hz, 14-H), 6.89 (1H, d, *J* = 8.3 Hz, 13-H), 6.35 (1H, d, *J* = 2.3 Hz, 6-H), 6.31 (1H, d, *J* = 2.3 Hz, 8-H), 5.68 (1H, m, -OH), 5.17–4.99 (8H, m, CH_2_Ph), 4.95 (1H, d, *J* = 11.7 Hz, 2-H), 4.79 (1H, d, *J* = 11.7 Hz, 3-H), 3.13 (2H, d, *J* = 3.6 Hz, 4-H_2_) ppm.

(2*R*,3*R*)-2-(3,4-Dihydroxyphenyl)-5,7-dihydroxychroman-3-yl benzoate (**13b**) was obtained as a white powder. Under a hydrogen atmosphere, Pd(OH)_2_ (palladium hydroxide) (30 mg, 20% on carbon) was added to a solution of compound **13a** (28 mg, 0.037 mmol) in a solvent mixture of THF (tetrahydrofuran)/MeOH (methanol) (1:1 *v/v*, 3 mL) under a hydrogen atmosphere. The reaction mixture was stirred at room temperature and was monitored by TLC. When the reaction was completed, the reaction mixture was filtered to remove the catalyst. The product **13b** (83%) was purified by PTLC (n-Hex/EA 1:1) to afford a white powder. ^1^H NMR (CD_3_OD, 400 MHz) *δ* 7.87 (2H, dd, *J* = 8.5, 1.2, 2′ and 6′-H_2_), 7.55–7.48 (1H, m, 5′-H), 7.39 (2H, dd, *J* = 7.5, 1.6, 3′ and 5′-Hs), 6.96 (1H, d, *J* = 2.0 Hz, 10-H), 6.79 (dd, *J* = 8.2,2.5, 14-H), 6.68 (1H, d, *J* = 8.2 Hz, 13-H), 5.99 (1H, d, *J* = 2.3 Hz, 6-H), 5.97 (1H, d, *J* = 2.3 Hz, 8-H), 5.58 (1H, m, 3-H), 5.08 (1H, d, *J* = 11.6 Hz, 2-H), 3.04 (1H, dd, *J* = 17.4, 4.6 Hz, 4β-H), 2.91 (1H, dd, *J* = 17.6, 2.6 Hz, 4α-H) ppm [51].

(2*R*,3*R*)-5,7-Bis(benzyloxy)-2-(3,4-bis(benzyloxy)phenyl)chroman-3-yl 4-fluorobenzoate (**14a**) was obtained as a yellow oily liquid. A solution of compound **12** (0.07 mmol) and 4-DMAP (0.175 mmol) in CH_2_Cl_2_ (4 mL) was added dropwisely to a solution of 4-fluorobenzoyl chloride (0.14 mmol) in CH_2_Cl_2_ (1 mL) at 0 °C. The reaction mixture was stirred at room temperature for 16 h. Saturated NaHCO_3_ solution was added into the reaction mixture, and the organic layer was separated. The aqueous layer was further extracted with CH_2_Cl_2_. The combined CH_2_Cl_2_ layer was then dried over Na_2_SO_4_. The dried and evaporated residue was purified over PTLC (n-Hex/EtOAc 3:1) to afford compound **14a** as a yellow oily liquid (98%). HRESIMS 773.2673 *m/z* [M + H]^+^ (calculated for C_50_H_41F_O_7_, 773.2914). ^1^H NMR (CDCl_3_, 400 MHz) *δ* 7.95 (2H, dd, *J* = 8.8, 5.4 Hz, H-2′ and H-6′), 7.42–7.28 (20 H, m, Ar-H), 7.13 (1H, d, *J* = 2.1 Hz, H-10′), 7.03 (2H, t, *J* = 8.7 Hz, H-3′), 6.98 (1H, dd, *J* = 8.3, 1.8 Hz, H-14), 6.88 (1H, d, *J* = 8.3 Hz, H-13), 6.33 (1H, d, *J* = 2.3 Hz, H-6), 6.30 (1H, d, *J* = 2.3 Hz, H-8), 5.65 (1H, m, -OH), 5.14–4.99 (8H, m CH_2_Ph), 4.97 (1H, d, *J* = 11.8 Hz, H-2), 4.84 (1H, d, *J* = 11.8 Hz, H-3), 3.10 (2H, t, *J* = 3.8, 3.8 Hz, H-4α and 4β) ppm. ^13^C NMR (100 MHz, CDCl_3_) *δ* 167.1 (C-4′), 164.7164.7 (C-15), 158.9 (C-5), 158.0 (C-8a), 155.7 (C-7), 149.0 (C-11), 148.9 (C-12), 137.2, 137.1, 136.9, 136.9 (4 CH, PhCH_2_), 132.5 (C-1′), 132.4, 131.1, 128.7, 128.7, 128.6, 128.6, 128.5, 128.1, 128.0, 127.9, 127.8, 127.7, 127.4, 127.3, 127.3, 126.3 (16 CH, PhCH_2_), 119.9 (C-14), 115.7 (C-7), 115.5 (C-3′ and C-6′), 114.8 (C-13), 113.7 (C-10), 100.9 (C-4a), 94.8 (C-8), 94.0 (C-6), 77.5 (C-2), 71.4, 71.3, 70.2, 70.0 (4 CH, PhCH_2_), 68.8 (C-3), 26.2 (C-4) ppm.

(2*R*,3*R*)-2-(3,4-Dihydroxyphenyl)-5,7-dihydroxychroman-3-yl 4-fluorobenzoate (**14b**). Under a hydrogen atmosphere, Pd(OH)_2_ (52 mg, 20% on carbon) was added to a solution of compound **14a** (40 mg, 0.052 mmol) in a solvent mixture of THF/MeOH (1:1 *v/v*, 4 mL) under a hydrogen atmosphere. The reaction mixture was stirred at room temperature and was monitored by TLC. When the reaction was completed, the reaction mixture was filtered to remove the catalyst. The product **14b** (52%) was purified by PTLC (n-hexane /EtOAc 1:1). HRESIMS 413.1036 *m/z* [M + H]^+^ (calculated for C_22_H_17_FO_7_, 413.1037). ^1^H NMR (acetone-*d*_6_, 400 MHz) *δ* 8.32 (1H, s, -OH), 8.13 (1H, s, -OH), 7.98–7.92 (2H, m, -OH), 7.87–7.81 (2H, m, H-2′ and H-6′), 7.23–7.17 (2H, m, H-3′ and H-5′), 7.08 (1H, d, *J* = 2.1 Hz, H-10), 6.89 (1H, dd, *J* = 8.3, 2.1 Hz, H-14), 6.76 (1H, d, *J* = 8.2 Hz, H-13), 6.07 (1H, d, *J* = 2.3 Hz, H-6), 6.05 (1H, d, *J* = 2.3 Hz, H-8), 55.62 (1H, m, H-3), 5.20–5.16 (1H, m, H-2), 3.10 (1H, dd, *J* = 17.0, 4.5 Hz, H-4β), 2.98 (1H, dd, *J* = 17.6, 2.6 Hz, H-4α) ppm. ^13^C NMR (100 MHz, acetone-*d*_6_) *δ* 165.1 (C-15), 157.9 (C-7), 157.5 (C-8a), 157.0 (C-5), 145.7 (C-12), 145.6 (C-11), 133.1 (C-6′), 133.0 (C-1′), 131.3 (C-9), 127.7 (C-4′), 118.9 (C-14), 116.5 (C-5′), 116.3 (C-3′), 115.7 (C-13), 114.7 (C-10), 98.7 (C-4a), 96.6 (C-6), 95.8 (C-8), 77.8 (C-2), 70.4 (C-3), 26.5 (C-4) ppm.

(2*R*,3*R*)-5,7-Bis(benzyloxy)-2-(3,4-bis(benzyloxy)phenyl)chroman-3-yl 4-methoxybenzoate (**15a**). A solution of compound **12** (0.07 mmol) and 4-DMAP (0.175 mmol) in CH_2_Cl_2_ (4 mL) was added dropwisely to a solution of 4-methyloxybenzoyl chloride (0.14 mmol) in CH_2_Cl_2_ (1 mL) at 0 °C. The reaction mixture was stirred at room temperature for 16 h. Saturated NaHCO_3_ solution was added into the reaction mixture, and the organic layer was separated. The aqueous layer was further extracted with CH_2_Cl_2_. The combined CH_2_Cl_2_ layer was then dried over Na_2_SO_4_. The dried and evaporated residue was purified over PTLC (n-Hex/EtOAc 3:1) to afford compound **15a** (49%). HRESIMS 785.2936 *m/z* [M + H]^+^ (calculated for C_51_H_44_O_8_, 785.3114). ^1^H NMR (CDCl_3_, 400 MHz) δ 7.96–7.91 (2H, m, Ar-H), 7.41–7.29 (20H, m, Ar-H), 7.16 (1H, d, *J* = 2.0 Hz, H-10), 6.96 (1H, dd, *J* = 8.4, 2.1 Hz, H-14), 6.87 (1H, d, *J* = 8.3 Hz, H-13), 6.86–6.82 (2H, m, Ar-H), 6.33 (1H, d, *J* = 2.3 Hz, H-6), 6.29 (1H, d, *J* = 2.4 Hz, H-8), 5.65 (1H, m, -OH), 5.10–5.00 (8H, m, CH_2_Ph), 4.92 (1H, d, *J* = 11.7 Hz, H-2), 4.74 (1H, d, *J* = 11.7 Hz, H-3), 3.77 (3H, s, -OCH_3_), 3.13–3.07 (2H, m, H-4α and H-4 β) ppm. ^13^C NMR (100 MHz, CDCl_3_) δ 165.4 (C-4′), 163.5 (C-15), 158.8 (C-5), 158.1 (C-8a), 155.8 (C-7), 149.1 (C-11), 148.9 (C-12), 137.3 (C, PhCH_2_), 137.2, 137.0, 136.9, 132.0, 131.3, 128.7, 128.6, 128.6, 128.5, 128.4, 128.1, 128.0, 127.8, 127.7, 127.5, 127.33, 127.28 (17 CH, PhCH_2_), 122.5 (C-1′), 120.1 (C-11), 114.8 (C-13), 113.7 (C-10), 101.1 (C-4a), 94.8 (C-8), 93.9 (C-6), 77.8 (C-2), 71.3, 70.2, 70.0 (3 CH_2_, CH_2_Ph), 68.2 (C-3), 55.5 (-OCH_3_), 26.3 (C-4) ppm.

(2*R*,3*R*)-2-(3,4-Dihydroxyphenyl)-5,7-dihydroxychroman-3-yl 4-methoxybenzoate (**15b**) was obtained as a white solid. Under a hydrogen atmosphere, Pd(OH)_2_ (10 mg, 20% on carbon) was added to a solution of compound **15a** (20 mg, 0.025 mmol) in a solvent mixture of THF/MeOH (1:1 *v/v*, 4 mL) under a hydrogen atmosphere. The reaction mixture was stirred at room temperature and was monitored by TLC. When the reaction was completed, the reaction mixture was filtered to remove the catalyst. The product **15b** was purified by PTLC (n-hexane/EtOAc 1:1) (50%) to afford a white solid. HRESIMS 425.1235 *m/z* [M + H]^+^ (calculated for C_23_H_20_O_8_, 425.1236). ^1^H NMR (Acetone-*d*_6_, 400 MHz) δ 8.25 (1H, s, -OH), 8.06 (1H, s, -OH), 7.83–7.87 (2H, m, -OH), 7.85 (2H, d, *J* = 9.0 Hz), 7.85 (2H, d, *J* = 9.0 Hz, H-2′ and H-6′), 7.08 (1H, d, *J* = 2.1 Hz, H-10), 6.95 (2H, d, *J* = 9.0 Hz, H-3′ and H-5′), 6.90 (1H, dd, *J* = 8.3, 2.0 Hz, H-14), 6.76 (1H, d, *J* = 8.1 Hz, H-13), 6.06 (1H, d, *J* = 2.3 Hz, H-6), 6.04 (1H, d, *J* = 2.3 Hz, H-8), 5.58 (1H, ddd, *J* = 4.3, 2.6, 1.4 Hz, H-3), 5.17 (1H, s, H-2), 3.84 (3H, s, -OCH_3_), 3.08 (2H, dd, *J* = 17.4, 4.6 Hz, H-4β), 2.97 (2H, dd, *J* = 17.6, 2.6 Hz, H-4α) ppm. ^13^C NMR (100 MHz, acetone-*d*_6_) δ 165.7 (C-4′), 164.4 (C-15), 157.8 (C-7), 157.4 (C-8a), 157.0 (C-5), 145.6 (C-12), 145.5 (C-11), 132.2 (C-2′ and C-6′), 131.3 (C-9), 123.4 (C-1′), 119.0 (C-14), 115.6 (C-13), 114.8 (C-3′ and C-5′), 114.5 (C-10), 98.8 (C-4a), 96.4 (C-6), 95.7 (C-8), 77.9 (C-2), 69.7 (C-3), 55.8 (-OCH_3_), 26.5 (C-4) ppm.

(2*R*,3*R*)-5,7-Bis(benzyloxy)-2-(3,4-bis(benzyloxy)phenyl)chroman-3-yl cyclopentanecarboxylate (**16a**) was obtained as a white solid. A solution of compound **12** (0.07 mmol) and 4-DMAP (0.175 mmol) in CH_2_Cl_2_ (4 mL) was added dropwisely to a solution of cyclopentanecarbonyl chloride (0.14 mmol) in CH_2_Cl_2_ (1 mL) at 0 °C. The reaction mixture was stirred at room temperature for 16 h. Saturated NaHCO_3_ solution was added into the reaction mixture, and the organic layer was separated. The aqueous layer was further extracted with CH_2_Cl_2_. The combined CH_2_Cl_2_ layer was then dried over Na_2_SO_4_. The dried and evaporated residue was purified over PTLC (n-Hex/EtOAc 3:1) to afford compound **16a** (95.7%) as a white solid. HRESIMS 747.3079 *m/z* [M + H]^+^ (calculated for C_49_H_46_O_7_, 747.3322). ^1^H NMR (CDCl_3_, 400 MHz) δ 7.47–7.32 (20H, m, Ar-H), 7.14 (1H, d, *J* = 2.0 Hz, H-10), 6.98 (1H, dd, *J* = 8.4, 2.0 Hz, H-14), 6.93 (1H, d, *J* = 8.3 Hz, H-13), 6.31 (1H, d, *J* = 2.3 Hz, H-6), 6.29 (1H, d, *J* = 2.3 Hz, H-8), 5.47–5.42 (1H, m, -OH), 5.17 (4H, d, *J* = 2.5 Hz, CH_2_Ph), 5.05–5.00 (5H, m, H-2 and CH_2_Ph), 3.02 (1H, dd, *J* = 17.6, 4.6 Hz, H-4α), 2.98 (1H, dd, *J* = 17.5, 2.5 Hz, H-4β), 2.57 (1H, tt, *J* = 8.6, 7.4 Hz, H-1′), 1.36–1.75 (8H, m, H-2′, H-3′, H-4′ and H-5′) ppm. ^13^C NMR (100 MHz, CDCl_3_) δ 176. (C-15)1, 158.7 (C-5), 158.0 (C-8a), 155.5 (C-7), 148.9 (C-11), 148.7 (C-12), 137.3 (q, PhCH_2_), 137.0 (C, PhCH_2_), 131.2 (C-9), 128.7, 128.6, 128.5, 128.5, 128.1, 128.0, 127.9, 127.8, 127.7, 127.5, 127.3, 127.2 (12 CH, PhCH_2_), 119.7 (C-14), 114.9 (C-13), 113.7 (C-10), 101.0 (C-4a), 94.7 (C-8), 93.9 (C-6), 77.2 (C-2), 71.6, 71.4, 70.2, 69.9 (CH_2_, CH_2_Ph), 43.8 (C-1′), 30.3 (C-5′), 29.7 (C-2′), 26.0 (C-4), 25.8 (C-4′), 25.7 (C-3′) ppm.

(2*R*,3*R*)-2-(3,4-Dihydroxyphenyl)-5,7-dihydroxychroman-3-yl cyclopentanecarboxylate (**16b**). Under a hydrogen atmosphere, Pd(OH)_2_ (10 mg, 20% on carbon) was added to a solution of compound **16a** (40 mg, 0.05 mmol) in a solvent mixture of THF/MeOH (1:1 *v/v*, 4 mL) under a hydrogen atmosphere. The reaction mixture was stirred at room temperature and was monitored by TLC. When the reaction was completed, the reaction mixture was filtered to remove the catalyst. The product **16b** was purified by PTLC (n-hexane/EtOAc 1:1) (50%). HRESIMS 387.1440 *m/z* [M + H]^+^ (calculated for C_21_H_22_O_7_, 387.1443). ^1^H NMR (acetone-*d*_6_, 400 MHz) δ 8.09 (4H, s, -OH), 7.02 (1H, d, *J* = 1.7 Hz, H-10), 6.86–6.76 (2H, m, H-13 and H-14), 6.06 (1H, d, *J* = 2.3 Hz, H-6), 5.97 (1H, d, *J* = 2.3 Hz, H-8), 5.37 (1H, ddd, *J* = 4.7, 2.5, 1.4 Hz, H-3), 5.05 (1H, d, *J* = 1.2 Hz, H-2), 2.94 (1H, dd, *J* = 17.5, 4.6 Hz, H-4β), 2.79 (1H, dd, *J* = 17.4, 2.5 Hz, H-4α), 2.60 (1H, tdd, *J* = 10.2, 10.2, 7.5, 5.7 Hz, H-1′), 1.78–1.52 (4H, m, -CH_2_), 1.52–1.43 (4H, m, -CH_2_) ppm. ^13^C NMR (100 MHz, acetone-*d*_6_) δ 176.0 (C-15), 157.8 (C-7), 157.4 (C-8a), 157.0 (C-5), 145.6 (C-12), 145.5 (C-11), 131.3 (C-9), 119.0. (C-14), 115.6 (C-13), 114.9 (C-10), 98.8 (C-4a), 96.4 (C-6), 95.8 (C-8), 77.7 (C-2), 68.9 (C-3), 44.5 (C-1′), 26.3 (C-3′ and C-4′), 26.3 (C-2′ and C-5′), 26.2 (C-4) ppm.

(2*R*,3*R*)-5,7-Bis(benzyloxy)-2-(3,4-bis(benzyloxy)phenyl)chroman-3-yl cyclopropanecarboxylate (**17a**) was obtained as a white solid [52]. A solution of **12** (0.07 mmol) and 4-DMAP (0.175 mmol) in CH_2_Cl_2_ (4 mL) was added dropwisely to a solution of cyclopropanecarbonyl chloride (0.14 mmol) in CH_2_Cl_2_ (1 mL) at 0 °C. The reaction mixture was stirred at room temperature for 16 h. Saturated NaHCO_3_ solution was added into the reaction mixture, and the organic layer was separated. The aqueous layer was further extracted with CH_2_Cl_2_. The combined CH_2_Cl_2_ layer was then dried over Na_2_SO_4_. The dried and evaporated residue was purified over PTLC (n-hexane/EtOAc 3:1) to afford compound **17a** (93.5%) as a white solid. HRESIMS 719.2710 *m/z* [M + H]^+^ (calculated for C_47_H_42_O_7_, 719.3008). ^1^H NMR (CDCl_3_, 400 MHz) δ 7.48–7.32 (20H, m, Ar-H), 7.17 (1H, d, *J* = 1.8 Hz, H-10′), 6.97 (1H, dd, *J* = 8.4, 1.8 Hz, H-14), 6.93 (1H, d, *J* = 8.4 Hz, H-13), 6.31 (1H, d, *J* = 2.2 Hz, H-6), 6.30 (1H, d, *J* = 2.3 Hz, H-8), 5.44 (1H, s, -OH), 5.22–5.14 (4H, m, CH_2_Ph), 5.06–4.97 (6H, m, H-2′, H-3′ and CH_2_Ph), 3.01 (1H, dd, *J* = 16.9, 4.5 Hz, H-4α), 2.99 (1H, dd, *J* = 16.9, 2.8 Hz, H-4β), 1.50 (1H, tt, *J* = 8.2, 4.7 Hz, H-1), 0.89–0.68 (4H, m, H-2′ and H-3′) ppm [53].

(2*R*,3*R*)-2-(3,4-Dihydroxyphenyl)-5,7-dihydroxychroman-3-yl cyclopropanecarboxylate (**17b**) was obtained as a white solid. Under a hydrogen atmosphere, Pd(OH)_2_ (40 mg, 20% on carbon) was added to a solution of compound **17a** (40 mg, 0.056 mmol) in a solvent mixture of THF/MeOH (1:1 *v/v*, 4 mL) under a hydrogen atmosphere. The reaction mixture was stirred at room temperature and was monitored by TLC. When the reaction was completed, the reaction mixture was filtered to remove the catalyst. The filtrate was evaporated and purified by PTLC (n-hexane/EtOAc 1:1) to afford compound **17b** (75%) as a white solid. ^1^H NMR (CD_3_OD, 400 MHz) δ 6.93 (1H, d, *J* = 1.4 Hz, H-10), 6.73–6.76 (2H, m, H-13 and H-14), 5.95 (1H, d, *J* = 2.3 Hz, H-7), 5.92 (1H, d, *J* = 2.3 Hz, H-6), 5.32 (1H, ddd, *J* = 4.7, 2.4, 1.4 Hz, H-3), 4.98–4.93 (1H, m, H-2), 2.91 (1H, dd, *J* = 17.4, 4.7 Hz, H-4α), 2.76 (1H, dd, *J* = 17.6, 2.4 Hz, H-4β), 1.60–1.48 (1H, m, H-1′), 0.90–0.65 (4H, m, H-2′and H-3′) ppm.

### 4.5. Bioactivity Evaluation Using EBOV Pseudotyped Virus

#### 4.5.1. Cell Cultures

293T (CRL-3216) and A549 (CRM-CCL-185) cells were obtained from the American Type Culture Collection (ATCC) and were grown in Dulbecco’s modified Eagle medium (DMEM, Gibco) containing glutamine, supplemented with 10% fetal bovine serum (FBS). The cells were cultured at 37 °C in a humidified incubator with 5% CO_2_. 

#### 4.5.2. Cytotoxicity Assay

A sulforhodamine B (SRB) assay was used to evaluate the cytotoxicity of the samples using A549 cells with modifications [54]. Briefly, 4000 cells in 190 μL complete DMEM medium were seeded in each well of a 96-well cell culture plate. After 24 h, 10 μL of a sample in 10% (*v/v*) DMSO was introduced to the 96-well plate. An amount of 10 μL of 10% (*v/v*) DMSO was used as negative control, and paclitaxol was used as a positive control. After the treated cells were incubated at 37 °C for 72 h, 50 μL cold trichloroacetic acid (TCA, 50%, *w*/*v*) was added into each well. The plates were incubated at 4 °C for an additional 1 h to fix the cells. The plates were then rinsed with low-running tap water four times and dried at room temperature. An amount of 50 μL of SRB solution (0.4%, wt/vol) was then added to each well. The plates were incubated at room temperature for 10 min and were washed with 1% (*v/v*) acetic acid to remove the unbound dye. The plates were allowed to dry. The protein-bound dye in each well was dissolved with 100 μL of Tris base solution (10 mM, pH 10.5), followed by shaking on an orbital shaker for at least 30 min. The optical density (OD) values were measured at 515 nm in a microplate reader. The CC_50_ (the concentration of an agent to cause a by 50% reduction in cell viability) values were calculated by GraphPad Prism 5.0 (GraphPad Software, San Diego, CA, USA).

#### 4.5.3. Production of EBOV and VSV Pseudovirions

Human embryonic kidney 293T cells were transiently transfected with either 1.875 µg VSV-G envelope-expression plasmid or 1.875 µg Ebola glycoprotein-expression plasmid (Zaire ebolavirus isolate IRF0164, partial genome) and 13.125 µg Env-deficient HIV vector (pNL4-3. Luc. R.E.) [55,56] in a 100 mm cell culture dish via Lipofectamine 3000 (ThermoFisher, Waltham, MA, USA) following the manufacturer’s instructions. This pNL4-3 was derived from an infectious molecular clone of an SI, T-tropic virus (Michael et al., 1998), and is replication-deficient since the HIV is Env- and Vpr-. Additionally, the luciferase gene (luc) carried by this recombinant HIV vector serves as the reporter for HIV replication (reverse transcription, integration, and HIV gene expression). Forty-eight hours post-transfection, the supernatants were collected and filtered through a 0.45 µm pore size filter, and the generated pseudovirions (EBOVpp (EBOV pseudovirion): HIV vector with EBOV-GP incorporated on the surface; VSVpp (VSV pseudovirion): HIV vector with VSV-GP incorporated on the surface) were used for infection [44].

#### 4.5.4. Pseudovirion Inhibition Screening Assay

The host A549 (human lung epithelial) cells were seeded at 4000 cells per well (96-well plate) in complete DMEM. Plant extracts, fractions, or compounds (10 µL) and the pseudovirus (90 µL) were incubated with host cells. Seventy-two hours post-infection, cell lysates were collected for luciferase assay using a Neolite Reporter gene assay system (PerkinElmer). The luciferase signals were measured in a Perkin Elmer Envision 2104 multilabel reader.

### 4.6. Antiviral Assay against Infectious Ebola Virus

Samples were serially diluted to a ratio of 1:10, with 100 μM being the highest concentration tested. For determination of cell viability with CellTiter-Glo, HeLa cells were incubated for 48 h with samples, after which time an equal volume of CellTiter-Glo reagent was added. Luminescence values were determined for control untreated cells and were used as a reference for sample-treated cells. The anti-infectious EBOV assay was performed following the published protocol by Cui at et al. [44].

To evaluate whether the samples inhibit infection, 40 μL of a sample was added to each well of VERO E6 cells. The EBOV inoculum was delivered in 10 μL (MOI = 1). After adding virus, the cells were incubated for 1 h at 37 °C, after which time they received another 50 μL of sample. After 48 h, cells were fixed with 10% formalin, and infection was measured. EBOV-infected cells were detected with human Ab KZ52. An anti-human secondary conjugated to Alexafluor 488 was used to visualize infected cells that were counterstained with DAPI. Wells receiving only virus and media were used to establish a “control infection.” Control infection rates varied between plates, from approximately 11% to 35% (*n* = 4 wells per plate). Given that a 35% infection rate is more common than what was seen in our group, the higher infection rate was used to determine experimental infection rates. Experimental values are expressed as a percentage of the control infection.

### 4.7. Molecular Docking Analysis

#### 4.7.1. Preparation of Ligands for Docking

3D structures of all the compounds used in the study were constructed using Chem3D software (Molecular Modeling and Analysis; Cambridge Soft Corporation, Cambridge, MA, USA). The default root, rotatable bonds, and torsions of the ligand were set by the TORSDOF utility in AutoDock Tools.

#### 4.7.2. Preparation of Proteins for Docking

The target protein, Ebola glycoprotein, was retrieved from the Protein Data Bank (https://www.rcsb.org/structure/5JQ7) (accessed on 30 December 2021). All bound waters, cofactors, and the original ligand were removed from the protein. Geisteger charges were computed, and polar hydrogen atoms were added subsequently. The AutoDock atom types were defined using AUTODOCK Tools 1.5.6, the graphical user interface of AUTODOCK, supplied by MGL Tools. The grid parameters were adjusted to cover the binding site of the original ligand in the protein crystal structure.

#### 4.7.3. Molecular Docking Using AutoDock Vina

The prepared pdbqt files of ligands and proteins were docked using AUTODOCK Vina [30]. For each docking, the docking poses with the eight best docking scores were obtained, and the binding interactions were analyzed.

### 4.8. Microscale Thermophoresis (MST) Experiment

The plasmids expressing His-tagged EBOV-GP (Ebola virus glycoprotein expression plasmid, C-His tag, cat. no.:VG40304-CH, Sino Biological) and the pCMV-negative control (cat. no.: CV015, Sino Biological) were transfected into 293T cells in a separate 90 mm dish using lipofectamine 3000. Forty-eight hours post-transfection, the transfected cells were washed by PBS twice, followed by the addition of M-PER lysis buffer supplemented with protease inhibitor cocktail (Halt ™ Protease Inhibitor Cocktail, EDTA-Free, Thermo Fisher Scientific) to collect the cell lysates.

MTS experiments were performed according to the NanoTemper Technologies protocol in a Monolith NT.115 (red/blue) instrument (NanoTemper Technologies, South San Francisco, CA, USA). In the experiment, the cell lysate expressing His-tagged proteins was labeled with a Monolith His-Tag Labeling Kit RED-tris-NTA (NanoTemper Technologies), as described by the manufacturer. The final concentration of target protein in the cell lysate was kept constant at 25 nM. PBST/M-PER (1:1) was used as the assay buffer. A small-molecule DMSO stock solution (10 µL) was diluted to a ratio of 1:10 with assay buffer and then diluted to a ratio of 1:1 in 10 µL assay buffer to make a 16-sample dilution series (final concentration ranging from 1.52 nM to 50 µM). Labelled cell lysate (10 µL) was added to the diluted drug solution, and the mixture was incubated for 20 min in the dark. The mixtures were then loaded to standard Monolith NT.115 capillaries. The experiments were performed on a Monolith NT.115 Red system using medium MST power and 30% LED power at 25°C. The data acquired with MO. Control (NanoTemper Technologies) were analyzed using MO. Affinity Analysis 3.0.5 (NanoTemper Technologies), and the *K*_d_ values were determined by fitting the data with the *K*_d_ model.

## 5. Conclusions

Ebola virus disease (EVD) is associated with fulminant hemorrhagic fever and high case mortality rates. Researchers are making efforts to develop anti-EBOV drugs. However, biosafety level 4 laboratories are required to handle EBOV, which constrains EBOV-related studies. A pseudotyped virion system that consists of envelope-deficient core from a virus, luciferase reporter genes, and envelope glycoproteins from another virus provides a convenient alternative platform for antiviral research. In the present study, we established a pseudotyped screening system involving two types of pseudovirions, EBOVpp and VSVpp, to evaluate the anti-EBOV activity of a sample. By comparing the inhibitory-effect difference of a sample against the two pseudovirions, an antiviral agent that specifically targets EBOV can be determined.

Natural resources are reservoirs for druggable compounds. The considerable resources of plants in the Lingnan region provide an abundant potential to discover lead molecules. We screened 500 plant extracts made from medicinal plants collected in this region, which led to the discovery of *M. perlarius* as an anti-EBOV lead. Prior phytochemical and pharmacological studies of *Maesa* plants mainly focused on triterpenoid saponins. Our bioassay-guided fractionation study of the methanol extract of the stems of *M. perlarius* resulted in the discovery of a number of flavan-3-ol oligomers as potent EBOV inhibitors. In particular, we identified B-type procyanidins belonging to a class of condensed tannins as a new category of anti-EBOV phytochemicals. Furthermore, procyanidin B2 (**1**) was docked to fit the binding pocket of EBOV-GP, which was confirmed in MTS experiments by comparing it with the known EBOV-GP ligand toremifene. Thus, the current study presents flavan-3-ol monomers and oligomers as lead molecules that can be used as scaffolds for a target-oriented synthesis of additional analogues possessing improved anti-EBOV potency.

## Data Availability

The data presented in this study are available on request from the corresponding authors.

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
