# Peer review of "Ebola Entry Inhibitors Discovered from Maesa perlarius"

_ijms, 2022, doi:10.3390/ijms23052620_

Round 1
Reviewer 1 Report
In my opinion, the manuscript would be more suitable for another journal.
In Introduction section, the Authors should try to make an effort to emphasize the importance of their studies. The aims of the study should be reformulated according to the research objectives and target results.
Discussion is very important part of each manuscript published. In presented paper this section is poor and comprises too general explanations.
The conclusions should be integrated with more detailed results summarizing all the study and must reflect the innovation of this study and the perspectives.
The Authors should also use the correct name of studied species.
What about statistical results?
English and style require a careful reorganization. There are a lot of language mistakes, grammatically and stylistically.
Author Response
Thanks for the reviewer's comments and suggestions. Please see the attached file for our responses.

Reviewer 2 Report
Dear authors, the manuscript, entitled "Ebola Entry Inhibitors Discovered from Maesa perlarius" present the description of novel antiviral compounds against Ebola virus. Several points can be adressed to strenghten the manuscript :
Line 73 : There is no experimental evidence of the mechanism of action of the compounds, then the word "elucidation" is overstated.
Line 93 : The term Ebola pseudovirus may be miss understood , pleas consider for the first statement, Ebola pseudotyped HIV particles (EBOV-pp)
Figure 1 : The method to calculate IC50 is not indicated
Figure2 : several compounds induce negative inhibition % in host cells, fraction C to G, there is no comment about it
Line 101 : It is indicated that the data suggest that the action is on entry stage. However it cannot be excluded that compounds prevent binding. In addition, the compound may have action, on internalization, digestion, receptor binding , fusion. There is no experiment to decipher the precise step, neither it is discussed.
Line 153 : check sentence
Line 186 : There is no clear reason why the molecules are docked in the toremifen binding cavity. As the mechanism of action is not defined, this region of the protein cannot be specificallytargeted.
Line 257-272 : This section is containing introduction details and results repetion, please move this paragrph in the right section
Conclusion : Line 632-639, this section contains material and methods (or introductive subjects) but not conclusive remarks
Author Response

(The authors gave the same response as above.)

Reviewer 3 Report
Remarks to the Author:
In the manuscript ‘Ebola Entry Inhibitors Discovered from Maesa perlarius’, Nga Yi Tsang et al. reported isolation and identification of chemical compounds from plants showing potent neutralizing capacity against pseudovirus carrying Ebola virus glycoprotein, which indicate their inhibitory effect on the virus entry process.
Major comments:
- line42, the description leading readers to learn about Lingnan is good, but from the perspective of molecular research, the introduction should be briefer
- line79, need to point out this is replication-incompetent or competent virus
- for Fig 1:
(1) should include at least two negative controls, one is HIV-Luc without insertion of Ebola GP, the other one is solvent without MP extraction
(2) keep the title of 1A x and y axis consistent with 1B, and all through the manuscript, either use % Inhibition or % Infection, which improves the readability
(3) the unit of x axis is ng/ml or ug/ml?
- same comments here as above, need to point out this is replication-incompetent virus
- line108, 'fractionation'
- line113, the IC50 value is lack of unit
- for Result 2.5., this computational evidence is too weak to make the point.
please provide results from, e.g., binding assay (like SPR) to show these small molecular compounds could bind GP, or competition assay to show these identified compounds target the similar region with toremifene as stated in this section
- line200, binding score or docking score?
- for Table 2:
(1) using scientific format for the IC50 value
(2) could remove pIC50 value from this table, not necessary to show this value here
- For Fig 7:
(1) the title of y axis should be docking score, not the software name
(2) please label p value in the graph
(3) what statistical correlation model was used for this analysis? need to point out or write down in the method section
(4) suggest putting Fig 7 into suppl figs
- the discussion section should be a deeper talk extended from the results, not just a literature review. Please add contents appropriately, such as discussing the potential molecular mechanism for their neut capability, and like how the analogues or derivatives get better neut.
Author Response

(The authors gave the same response as above.)

Round 2
Reviewer 2 Report
thanks for incorporating remarks
Reviewer 3 Report
Thanks for the authors' response!